# n-Butylidenephthalide recovered calcium homeostasis to ameliorate neurodegeneration of motor neurons derived from amyotrophic lateral sclerosis iPSCs

Yu-Chen Deng[1,2], Jen-Wei Liu[2], Hsiao-Chien Ting[3], Tzu-Chen Kuo[1], Chia-Hung Chiang[1], En-Yi Lin[1,3], Horng-Jyh Harn[3,4], Shinn-Zong Lin[3,5], Chia-Yu Chang[3,6,7]*, Tzyy-Wen Chiou[1]*

1 Department of Biochemical and Molecular Medical Sciences, National Dong Hwa University, Hualien, Taiwan, 2 Everfront Biotech Inc., Taipei, Taiwan, 3 Bioinnovation Center, Buddhist Tzu Chi Medical Foundation, Hualien, Taiwan, 4 Department of Pathology, Hualien Tzu Chi Hospital, Hualien, Taiwan, 5 Department of Neurosurgery, Hualien Tzu Chi Hospital, Hualien, Taiwan, 6 Department of Nursing, Tzu Chi University of Science and Technology, Hualien, Taiwan, 7 Department of Medical Research, Hualien Tzu Chi Hospital, Hualien, Taiwan

* twchiou@gms.ndhu.edu.tw (T-WC); scata0726@gmail.com (C-YC)

**Data Availability Statement:** All relevant data are within the manuscript and its Supporting Information files.

## Abstract

Amyotrophic lateral sclerosis (ALS) is an incurable neurodegenerative disease that causes muscle atrophy and primarily targets motor neurons (MNs). Approximately 20% of familial ALS cases are caused by gain-of-function mutations in superoxide dismutase 1 (SOD1), leading to MN degeneration and ion channel dysfunction. Previous studies have shown that n-Butylidenephthalide (BP) delays disease progression and prolongs survival in animal models of ALS. However, no studies have been conducted on models from human sources. Herein, we examined the protective efficacy of BP on MNs derived from induced pluripotent stem cells (iPSCs) of an ALS patient harboring the $SOD1^{G85R}$ mutation as well as on those derived from genetically corrected iPSCs ($SOD1^{G85G}$). Our results demonstrated that the motor neurons differentiated from iPSC with $SOD1^{G85R}$ mutation exhibited characteristics of neuron degeneration (as indicated by the reduction of neurofilament expression) and ion channel dysfunction (in response to potassium chloride (KCl) and L-glutamate stimulation), in contrast to those derived from the gene corrected iPSC ($SOD1^{G85G}$). Meanwhile, BP treatment effectively restored calcium ion channel function by reducing the expression of glutamate receptors including glutamate ionotropic receptor AMPA type subunit 3 (GluR3) and glutamate ionotropic receptor NMDA type subunit 1 (NMDAR1). Additionally, BP treatment activated autophagic pathway to attenuate neuron degeneration. Overall, this study supports the therapeutic effects of BP on ALS patient-derived neuron cells, and suggests that BP may be a promising candidate for future drug development.

**Funding:** This research was funded by the following grants: 1. Funder: National Science and Technology Council of Taiwan Award Number: MOST 111-2314-B-303-027-MY2 and NSTC 113-2314-B-303-026 Recipient: Chia-Yu Chang 2. Funder: Hualien Tzu Chi Hospital Award Number: IMAR-110-01-20 Recipient: Chia-Yu Chang The funders had no role in the study design, data collection, analysis, or decision to publish.

**Competing interests:** The authors declare that they have no competing interests.

## Introduction

Amyotrophic lateral sclerosis (ALS) is one of the most fatal motor neuron (MN) degenerative diseases characterized by the progressive loss of MNs in the spinal cord and brain [1,2]. Within 3–5 years of diagnosis, the disease destroys the MNs that control voluntary muscle movement, causing paralysis and death [3,4]. The estimated global prevalence and incidence of ALS are 4.42 per 100,000 and 1.59 per 100,000, respectively, and there is currently no effective treatment established for it [5].

Approximately 90% of ALS cases are sporadic, and approximately 10% are familial. Of these familial cases, 20% result from mutations in the gene encoding superoxide dismutase 1 (SOD1), an antioxidant enzyme that accounts for 1%–2% of the total soluble protein in the central nervous system (CNS) [6]. Wild type (WT) SOD1 protects neurons against oxidative stress from excessive reactive oxygen species (ROS) generation by converting superoxide anions into oxygen and hydrogen peroxide, whereas SOD1 mutants associated with neurodegenerative disease induce cell death through functional gain, although the exact pathological mechanism remains unclear [1]. Mice expressing the SOD1$^{G85R}$ mutation are a widely studied ALS disease model as the G85R substitution induces protein misfolding and aggregation, yielding inclusion bodies that cause MN neurotoxicity resembling that of the human disorder [7,8]. It is believed that the SOD1$^{G85R}$ variant influences copper and zinc binding characteristics at the catalytic site, allowing these metal ions to adversely influence other cellular processes, and conferring other conformational changes that promote protein aggregation and dysfunction [8,9].

There are many factors considered to be the cause of motor neuronal cell death in ALS. The clinical pathology of ALS includes neurofilament (NF) degeneration and loss of essential cytoskeletal functions in MNs [10,11]. Induced pluripotent stem cells (iPSCs) from familial ALS patients produce MNs in culture with reduced neurite length and limited survival [1] due in part to a greater apoptosis rate [12] compared to MNs derived from WT iPSCs. In addition to cytoskeletal dysfunction, MNs harboring ALS-associated mutations exhibit impaired calcium homeostasis, leading to changes in excitability, failure of action potential conduction, and activation of various calcium-dependent degenerative pathways [13–15]. For example, excessive presynaptic glutamate release and overactivation of postsynaptic glutamate receptors result in pathological intracellular calcium accumulation known as excitotoxicity [16,17].

The α-amino-3-hydroxy-5-methyl-4-isoxazolepropionic acid receptor (AMPAR) and N-methyl-D-aspartate receptor (NMDAR) are the primary glutamate receptor subtypes mediating fast excitatory synaptic transmission in the mammalian CNS [18]. These receptors also modulate synaptic strength, especially the NMDAR [19]. The AMPAR is a heteromultimer consisting of various combinations of GluR1, GluR2, GluR3, and GluR4 subunits. Channels consisting of GluR1, GluR3, and GluR4 are permeable to calcium in addition to sodium and potassium, while GluR2 blocks or permits calcium ion impermeability depending on a Q/R polymorphism. Most AMPARs include GluR2, thus the Q/R status frequently determines the calcium permeability. Overexpression of the calcium-impermeable variant in ALS SOD1 transgenic mice reduces MN calcium influx, promotes MN survival, and delays the onset of disease [20,21]. The GluR3 subunit also confers calcium permeability, and expression levels are upregulated at both mRNA and protein levels in SOD1$^{G93A}$ mutant mice, causing excessive calcium influx and glutamate excitotoxicity [22,23]. However, NMDARs have the greatest calcium permeability among the ionotropic glutamate receptors, and hyperactivation is a major inducer of calcium-dependent neuronal death [24]. These receptors are also heterotetramers, in this case composed of one or more obligatory NR1 subunits plus NR2A–D and (in some cases) NR3A/B. Adding to this complexity, these subunits have alternatively spliced variants

[25]. Marcuzzo and colleagues (2019) reported that a SOD1$^{G93A}$ transgenic mouse exhibited significantly higher levels of NR1 protein and mRNA expression compared to WT mice, while Fuller and colleagues (2006) reported that NR1 is overexpressed in motor nuclei, suggesting contributions to MN loss [26,27]. Consistent with this notion, drugs that can antagonize glutamate receptor activation such as Riluzole can delay the progression of ALS [28]. The glutamate receptor antagonists ketamine, amantadine, memantine, and dextromethorphan are also approved by the United States Food and Drug Administration (FDA) for various indications, including neurodegenerative diseases. Thus, targeting the glutamate receptor may prevent intracellular calcium dysregulation and ensuing excitotoxicity in ALS.

n-Butylidenephthalide (BP) is among the main components extracted from *Angelica sinensis* oil, an herb used in traditional Chinese medicine for the treatment angina and atherosclerosis, antagonizes numerous common pathogenic processes such as platelet agglutination, inflammation, and apoptosis [29–33]. Furthermore, BP inhibits autophagic activity, inflammation, and oxidative stress to reduce MN loss and prolong the survival of ALS mice [34]. Additionally, BP effectively reduces β-amyloid accumulation and improve short-term memory in an Alzheimer's disease (AD) mouse model by modulating the LncCYP3A43-2/miR29-2-5p/PSEN1 network [35]. Furthermore, BP reduces neuron cell loss in an animal model of spinocerebellar ataxia (SCA) by regulating autophagy, and reduce the excitotoxicity of ataxin-3 (ATXN3) on Purkinje progenitor cells from SCA3 patient-derived iPSCs by inhibiting the activity of the calcium-activated proteolytic enzyme calpain [36,37].

Although genetically modified animals and cell lines have provided crucial insights into ALS pathogenesis, there are clinically significant anatomical, physiological, and genetic differences between humans and animals that influence ALS pathogenesis and symptom progression [38]. Moreover, there are over 30 familial ALS subtypes, each with a distinctive molecular pathology and constellation of clinical phenotypes, and preclinical models do not fully recapitulate key pathological pathways [39]. Specifically, mutated SOD1 transgenic mice usually require 20–40-fold protein overexpression to exhibit symptoms [40], whereas only a single mutated gene copy confers familial ALS. Moreover, the pathology and sequence induced by mutant overexpression may be distinct despite similar symptoms [41]. In recent years, patient-derived iPSCs have contributed to the development of drugs for neurodegenerative disease treatment [42]. These cells can be applied for cell therapy and research on specific patients without the ethical issues associated with fetal cells, thereby increasing the probability of successful translation from research to clinical trials. In the current study, we used human iPSCs (SOD1$^{G85R}$) from ALS patients to derive MNs, and examined the efficacy of BP to improve stability of neurofilament, calcium homeostasis, and ion channel expression in iPSC-derived MNs.

## Methods

### Human-induced pluripotent stem cell culture

This study used two iPSC lines (SOD1$^{G85R}$ and SOD1$^{G85G}$) derived from ALS patient provided by Dr. C.Y. Chang (Buddhist Tzu Chi Medical Foundation). Mutant iPSCs were isolated from the peripheral blood mononuclear cells of ALS-related patient through reprograming and clonal selection, and CRISPR/Cas9 gene editing was used to correct the mutation for the control cell line SOD1$^{G85G}$ [3]. Both iPSC populations were incubated in Essential 8™ medium (Thermo Fisher Scientific Waltham, MA, USA), subcultured for approximately 3–4 days until 70%–80% confluent, and detached cells with accutase (Merck-Millipore), subsequently reseeded on plates precoated with Geltrex (Thermo Fisher Scientific). The rho-associated protein kinase (ROCK) inhibitor Y27632 was then added to enhance progenitor cell survival.

## MNs differentiation and characterization

The MN differentiation procedure used here was based on Ting's report [3]. Briefly, 1 mg/ml dispase II (Sigma-Aldrich) was added to iPSC cultures at 80% confluence for 5–10 min until cell colonies rounded and were less adherent. These colonies were then scraped off and incubated in petri dishes (Alpha Plus) containing Essential 6™ medium (Thermo Fisher Scientific) and refreshed 4–6 h after seeding. Subsequently, cultures were incubated for one day to form embryoid bodies (EBs), and then incubated in neural induction medium consisting of DMEM-F12 and DMEM/HG with 1% N-2 Supplement, 1 mM nonessential amino acids (NEAAs), 2 mM glutamine (all procured from Thermo Fisher Scientific), 10 μM SB431542 (Sigma-Aldrich), 10 ng/mL recombinant human FGF-basic (Peprotech), and 3 μM CHIR99021 (Cayman). After 4 days in induction medium, cells were cultured in Neurobasal™ Medium with 1% N-2 Supplement, 2 mM L-glutamine, 1 mM NEAAs, 2 μM SB431542 (Sigma-Aldrich), 0.2 μM LDN193189 (Cayman), 3 μM CHIR99021, 0.5 μM SAG (Cayman), and 0.5 μM retinoic acid (Sigma-Aldrich) to promote neuronal differentiation. During the differentiation period, various biomarkers were analyzed by immunofluorescence staining, as detailed in the next section, with DAPI (Thermo Fisher Scientific) counterstaining to assess differentiation status. Cells were immunostained for Olig2 (Novus Biologicals, Littleton, CO, USA) and Sox1 (R&D Systems, Minneapolis, MN, USA) on day 15 of differentiation, and for HB9 (DSHB) and NF-H (Sigma-Aldrich) on day 28–30. The HB9/DAPI ratio was calculated using ImageJ to quantify the final proportion of differentiated MNs.

## Immunofluorescence staining and quantification

Cells seeded on cover glasses (Deckgläser, Thermo Fisher Scientific) were washed with PBS, fixed with 4% paraformaldehyde (Sigma-Aldrich) for 15 min at room temperature (RT); washed three times with PBS; incubated on ice with 0.3% Triton-100 (J. T. Baker, Phillipsburg, NJ, USA) for 15 min; blocked using 5% horse serum (Thermo Fisher) at RT for 1 h; and incubated in antibodies against Olig2 (1:500), Sox1 (1:200), HB9 (1:100), and NF-H (1:1000) at 4°C for 16 h. Immunostained cultures were washed with PBST and incubated with Alexa Fluor-conjugated secondary antibodies (donkey anti-mouse IgG (H+L)-Alexa Fluor™ 488, donkey anti-rabbit IgG (H+L)-Alexa Fluor™ 594, donkey anti-goat IgG (H+L)-Alexa Fluor 594, and goat anti-rabbit IgG (H+L)-Alexa Fluor 546; all procured from Thermo Fisher) at RT for 1 h under darkness. Finally, cells were washed with PBST, counterstained with DAPI, and imaged through fluorescence microscopy (Imager Z1, Zeiss, Oberkochen, Germany). Both MN differentiation and neurite area were quantified using ImageJ.

## Calcium imaging

Cells seeded on coverslips (Deckgläser) were rinsed with physiological buffer (130 mM NaCl, 5 mM KCl, 2 mM $CaCl_2$, 10 mM HEPES, pH 7.4) containing 10 mM glucose, and incubated with 5 μM Fluo-4 (Thermo Fisher) at 37°C for 40 min under darkness. Then, Fluo-4-loaded cells were transferred to a calcium imaging chamber and washed with physiological buffer under darkness for 20 min. The ratio of $\triangle F/F_0$ was then measured during the indicated treatments for estimation of intracellular calcium. For these experiments, Fluo-4 loaded cells were treated with 60 mM KCl for 30 s, reperfused with physiological buffer for 300 s, and treated with 1 mM L-glutamate for 30 s. On the other hand, using agonist (10 μM S-AMPA, 40 μM NMDA and 10 μM S-AMPA + 40 μM NMDA) to treat Fluo-4-loaded cells for 30 s, and the cells will be washed with physiological buffer for 300 s between each agonist treatment. Fluo-4 emission ratio images were recorded using an ECLIPSE Ti2-E microscope (Nikon, Tokyo, Japan) and analyzed using NIS-Elements AR (Nikon).

## Western blotting analysis

Total protein was extracted from cells using RIPA lysis reagent (iNtRON Biotechnology) containing protease inhibitor (Thermo Fisher), separated by 12.5% sodium dodecyl sulfate-polyacrylamide gel electrophoresis (SDS-PAGE), and transferred to PVDF membranes (Merk Millipore) at 4˚C for 18 h. Subsequently, membranes were blocked in TBST containing 5% FBS (Thermo Fisher) at RT for 1 hour, and then probed with primary antibodies against GluR3 (GeneTex), NR1 (GeneTex), LC3BII (Abcam), p62 (Abcam), activated/ cleaved caspase-3 (GeneTex), and GAPDH (Merk Millipore) in TBST containing 3% FBS at 4˚C for 16 h. Membranes were washed three times with TBST at RT (10 min each), incubated with mouse and rabbit secondary antibodies (Merk Millipore) at RT for 1 h, rewashed three times in PBS, and incubated with horseradish peroxidase chemiluminescent substrate (Merk Millipore). Finally, signals were detected using the iBright CL1000 Imaging System (Thermo Fisher Scientific).

## Treatment of BP on MNs

After 35 days of differentiation, SOD1$^{G85R}$ and SOD1$^{G85G}$ MNs were treated with 0, 10, or 20 μM BP (Alfa Aesar) for 72 h (according to effective dose reported previously in an AD iPSC model [29]). BP was dissolved in DMSO (Sigma-Aldrich) as the 10% (w/v) BP/DMSO stock. Following the BP treatment, we conducted immunofluorescence staining, calcium imaging, and Western blot analysis to test efficacy and investigate mechanisms.

## Statistical analysis

Results are presented as the mean ± standard deviation (SD) of at least three independent experiments. Statistical differences between groups were tested using one-way ANOVA with Dunnet's test. For the comparison between only two groups (Figs 1E and 2C), a Student's t-test was used. p-value of $<0.05$ was considered statistically significant for all tests.

## Results

### The MN Differentiation ratio of both SOD1$^{G85R}$ iPSCs and SOD1$^{G85G}$ iPSCs exceeded 85%

SOD1$^{G85R}$ iPSCs and SOD1$^{G85G}$ iPSCs (genetically corrected controls) were established. The neuron differentiation protocol adapted from Ting et al. (2021) was performed [3] (Fig 1A). As depicted in Fig 1B, morphology observation in sequential neural differentiation was performed. On the first day (D0, defined as the EB stage), iPSCs were seeded in Essential 6™ medium, which resulted in the formation of cell clusters with the characteristics of ectoderm, mesoderm, and endoderm (embryoid bodies, EBs). On the following day, EBs were cultured in neuron induction medium for 4 days (days D1 to D4), during which neuroepithelial cells formed a tighter and denser structure around the cell sphere. These neuronal progenitors were then grown in Neurobasal™ Medium to promote MN differentiation. By day 15 of the differentiation period (D15), cells were significantly enlarged and hypertrophic, and exhibited a more compact structure by D28.

To confirm the progression to neuronal differentiation, cultures were analyzed at each stage of the process (EB formation, induction, differentiation) [43]. The cultures were immunostained for SOX1 and Olig2 on D15, and both were found to be widely expressed in SOD1$^{G85R}$ and SOD1$^{G85G}$ iPSC-derived progenitor cells (Fig 1C and 1D). Subsequently, cultures were stained for the MN-specific homeobox transcription factor HB9 (Fig 1C and 1D). Calculation of the HB9-positive to DAPI-positive ratio indicated that 85.36% ± 1.290% of

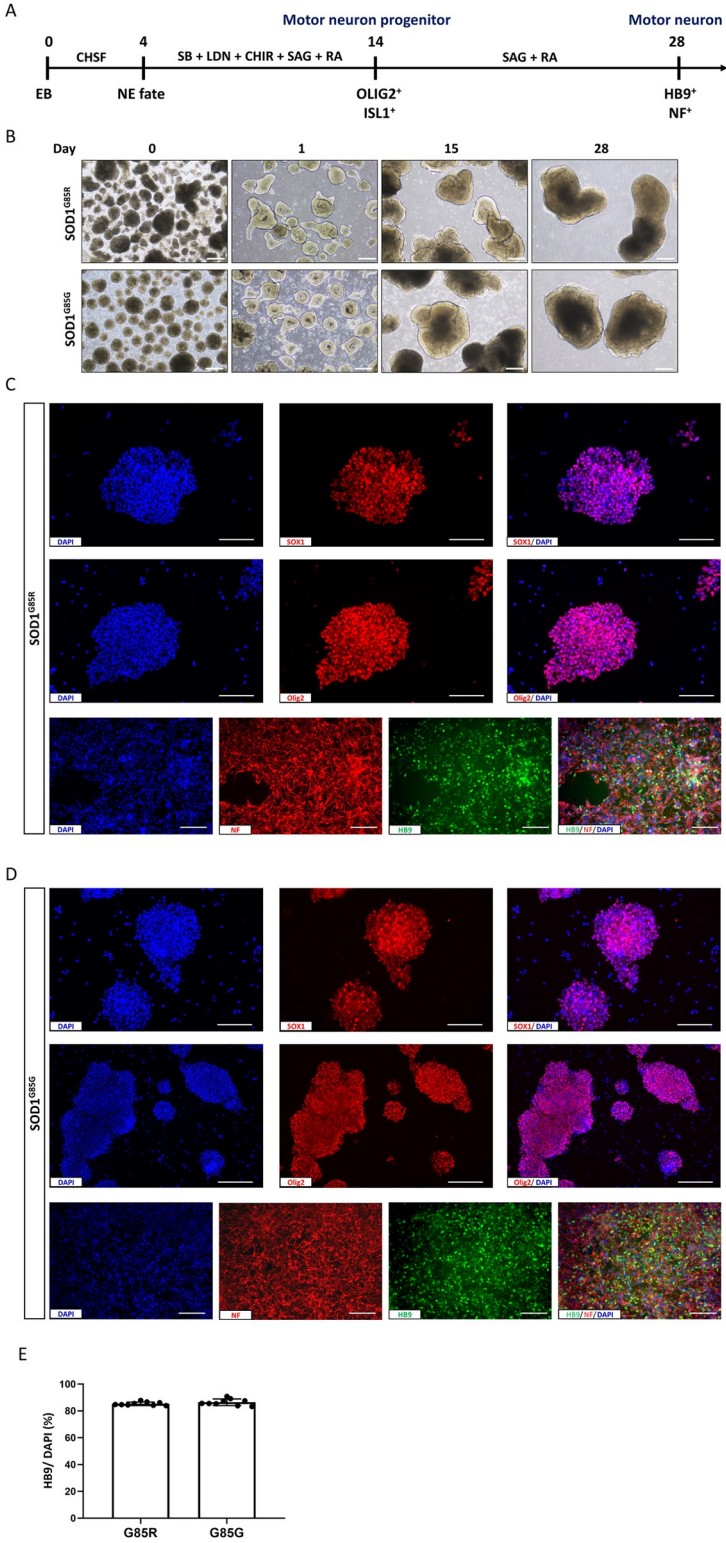

**Fig 1. Differentiation of human motor neurons (MNs) from the induced pluripotent stem cells (iPSCs) of an amyotrophic lateral sclerosis (ALS) patient.** (A) Schematic depicting the MN differentiation protocol. (B) Images showing changes in the morphology of MNs derived from SOD1$^{G85R}$ iPSCs (mutant) and SOD1$^{G85G}$ iPSCs (genetically corrected controls) during the differentiation process; scale bar, 300 μm. (C and D) Phenotype characterization of SOD1$^{G85R}$ and SOD1$^{G85G}$ iPSC-derived cells on days 15 and 30 of differentiation by

immunostaining for the MN progenitor markers SOX1 and Olig2 (red) and the mature MN markers (HB9, green) and NF-H (red), all with DAPI counterstaining (blue). (E) Quantification of the HB9-positive to DAPI (total cell) ratio. Mean ± SD from n = 3 independently treated cultures per treatment group; scale bar, 200 μm.

SOD1$^{G85R}$ iPSC-derived cells (ALS) and 86.51% ± 2.433% of SOD1$^{G85G}$ iPSC-derived cells (controls) were MNs by D28 in differentiation culture (Fig 1E). It revealed no significant difference between MNs from SOD1$^{G85R}$ iPSCs and SOD1$^{G85G}$ iPSCs. Thus, mutation in SOD1$^{G85R}$ did not influence that capacity of iPSCs to differentiate into MNs.

## Treatment of SOD1$^{G85R}$ iPSC-derived MNs with BP prevented glutamate-induced excitotoxicity

Calcium dysregulation and intracellular overload is a major contributor of progressive MN loss in ALS [15,44]. Fluo-4-based $Ca^{2+}$ microfluorometry was performed on MNs derived from SOD1$^{G85R}$-mutated iPSCs (G85R) and genetically corrected SOD1$^{G85G}$ iPSCs (G85G). The spontaneous calcium transients of MNs in the G85R and G85G were shown in Fig 2A and 2B. The results showed no significant difference in basal calcium content and spontaneous calcium transients between SOD1$^{G85R}$-derived (ALS) and SOD1$^{G85G}$-derived (control) groups (Fig 2C, G85R:0.196 ± 0.1251, G85G: 0.154 ± 0.1035). To examine whether BP can improve calcium homeostasis in ALS patient-derived MNs, we compared intracellular calcium responses among untreated and BP-treated SOD1$^{G85R}$-derived (ALS) and SOD1$^{G85G}$-derived (control). Furthermore, cells were treated with 60 mM KCl to stimulate voltage-gated calcium channels, yielding $\triangle F/F_0$ intensity ratios (positively correlated with intracellular calcium concentration) of 0.518 ± 0.3356 in untreated SOD1$^{G85R}$ MNs, 1.063 ± 1.4433 in BP-treated SOD1$^{G85R}$ MNs, and 1.258 ± 0.8873 in SOD1$^{G85G}$ MNs (Fig 2D–2F). Thus, voltage-gated calcium influx was 2.4-fold lower in SOD1$^{G85R}$ (ALS model) MNs compared to genetically corrected controls, indicating dysfunctional voltage-gated calcium channels or excitability mechanisms in ALS. However, voltage-gated calcium influx was restored by BP treatment (Fig 2G). Conversely, stimulation with L-glutamate evoked substantially greater calcium influx into untreated SOD1$^{G85R}$ MNs ($\triangle F/F_0$ = 3.130 ± 1.3304) compared to BP-treated SOD1$^{G85R}$ MNs and control SOD1$^{G85G}$ MNs (-0.044 ± 0.3724 and 0.265 ± 0.1904, respectively). Thus, while voltage-gated calcium influx was deficient, glutamate-induced influx was excessive in SOD1$^{G85R}$ MNs, and this abnormality was recovered by BP (Fig 2D–2H). These findings revealed that BP treatment can restore voltage-gated influx and prevent glutamate-induced excitotoxicity in MNs carrying an ALS-associated SOD1$^{G85R}$ mutation.

## BP downregulated glutamate receptor overexpression in SOD1 mutant MNs

Calcium influx induced by glutamatergic synaptic transmission is mediated mainly by AMPARs and NMDARs, and overactivation can cause selective death of MNs [45]. Therefore, we investigated whether BP (10 and 20 μM) could improve calcium ion overload by regulating the expression of AMPARs and NMDARs. The Western blotting results revealed that BP reduced NMDAR1expression (Fig 3A and 3B) and GluR3 expression (Fig 3C and 3D). In the 20 μM BP treatment group, expression levels of GluR3 were reduced to the level of SOD1$^{G85G}$ MNs. BP treatment reversed the overexpression of GluR3 (an AMPAR subunit) and NMDAR1 (an NMDAR subunit). This downregulation may explain how BP mitigated excessive glutamate-induced calcium influx and ameliorates excitotoxicity.

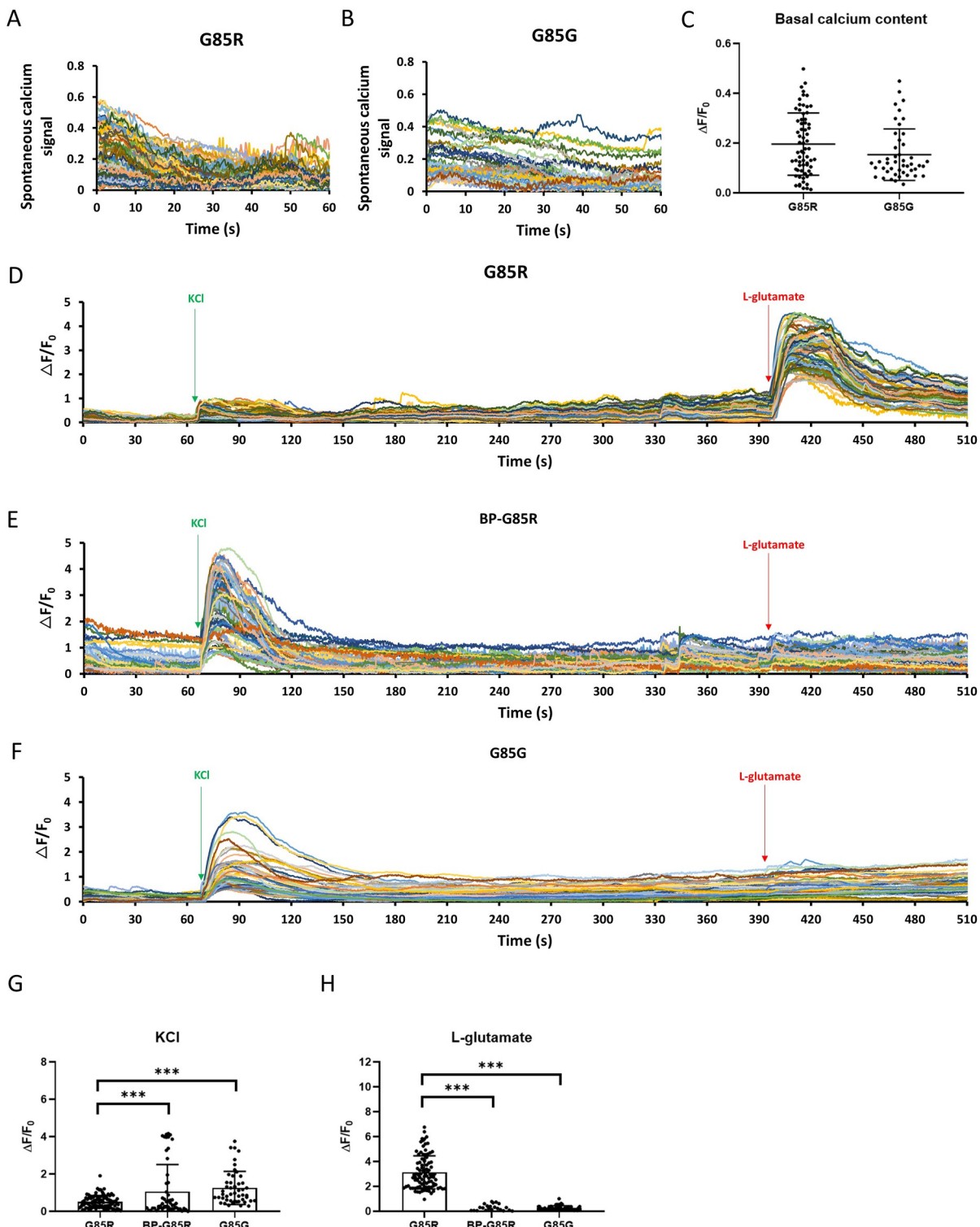

**Fig 2. Treatment with BP restores voltage-gated calcium channel function and reduces excessive calcium influx in ALS-SOD1 mutant MNs.** (A, B and C) Analysis of basal intracellular calcium levels and spontaneous calcium transients in untreated SOD1[G85R] iPSC-derived MNs (n = 67) (A and C) and SOD1[G85G] iPSC-derived MNs (controls, n = 46) (B and C). (D, E and F) Microfluorometric measurements of intracellular calcium transients in untreated SOD1[G85R] iPSC-derived MNs (D), SOD1[G85R] iPSC-derived MNs treated with 10 μM BP (E), and SOD1[G85G] iPSC-derived MNs (controls) (F) loaded with Fluo-4 and exposed to 60 mM KCl at 60 s and 1 mM L-glutamate at 390 s. (G and H) Average $\triangle F/F_0$ ratios during peak calcium induced by 60 mM KCl (G) and 1 mM L-glutamate (H). n = 95, 55 and 49 MNs for G85R, BP-G85R and G85G group, respectively. ***p < 0.005 vs. the untreated SOD1[G85R] group.

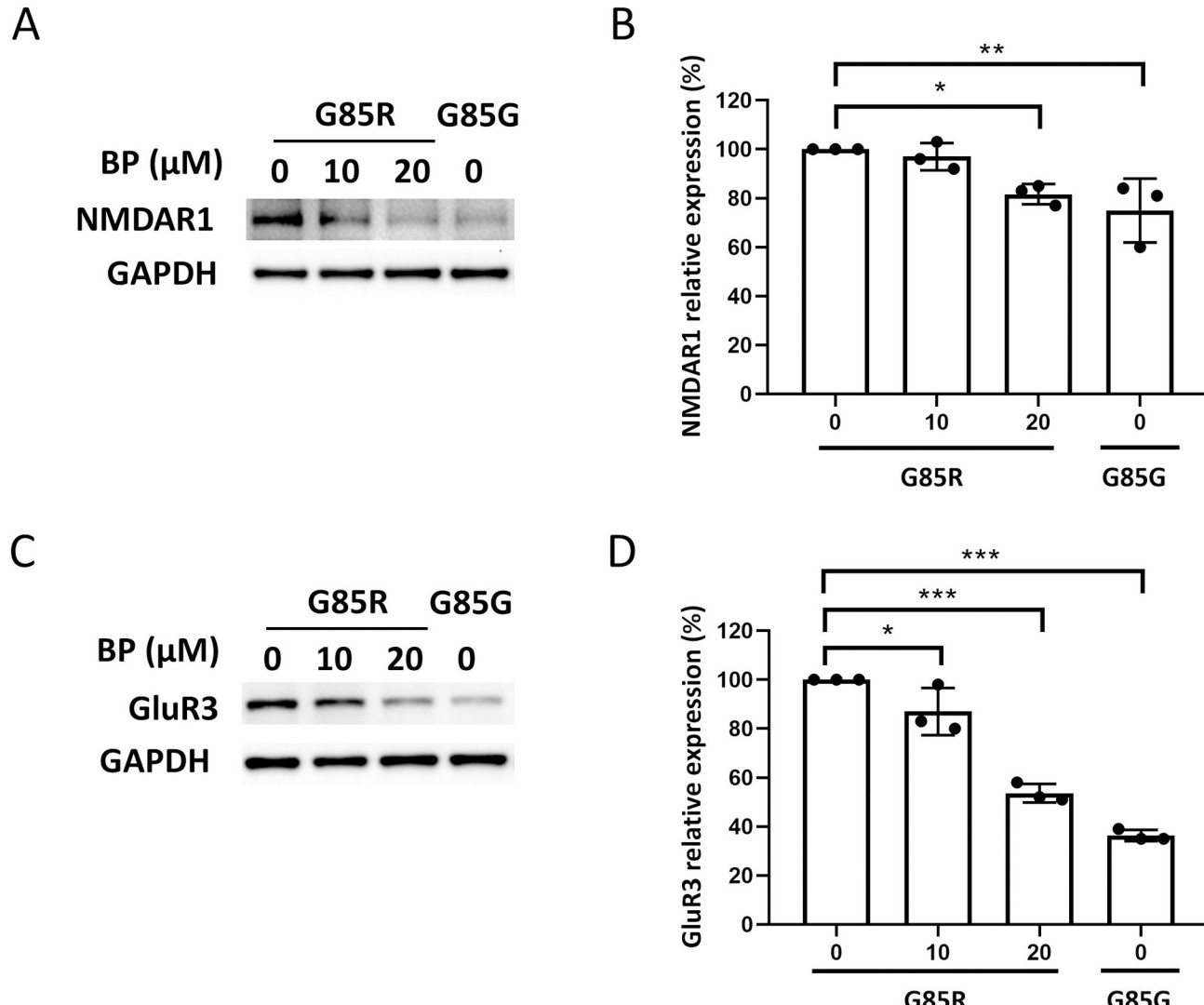

**Fig 3. Treatment with BP inhibits AMPAR and NMDAR glutamate receptor overexpression in ALS-SOD1 mutant MNs.** (A and C) Western blotting for NMDAR-NR1 (A) and AMPAR-GluR3 (C) expression in untreated SOD1$^{G85R}$ iPSC-derived MNs, SOD1$^{G85R}$ iPSC-derived MNs treated with BP (10 and 20 μM), and SOD1$^{G85G}$ iPSC-derived MNs (controls). (B and D) Quantification of the signal was performed using ImageJ.

## Treatment of SOD1$^{G85R}$ MNs with BP suppressed AMPAR and NMDAR agonist-induced excitotoxicity

The effects of BP in preventing glutamate-induced excitotoxicity and downregulating glutamate receptor overexpression in SOD1$^{G85R}$ MNs were confirmed in this study. We also examined the influence of BP on calcium flux through specific glutamate receptors by stimulating them with receptor-specific agonists, including S-AMPA (an AMPAR-specific agonist) and NMDA (an NMDAR-specific agonist). Cells treated with S-AMPA, NMDA, or S-AMPA + NMDA were examined through calcium imaging, which showed that SOD1$^{G85R}$ MNs exhibited stronger calcium flux in response to all agonist stimulations (S-AMPA: 0.491 ± 0.4262, NMDA: 0.147 ± 0.1312; S-AMPA + NMDA: 0.596 ± 0.5328) (Fig 4A) compared to SOD1$^{G85G}$ MNs (S-AMPA: 0.044 ± 0.0724; NMDA: 0.093 ± 0.0508; S-AMPA+NMDA: 0.017 ± 0.0888) (Fig 4C). Furthermore, after BP treatment, the calcium flux in SOD1$^{G85R}$ MNs decreased,

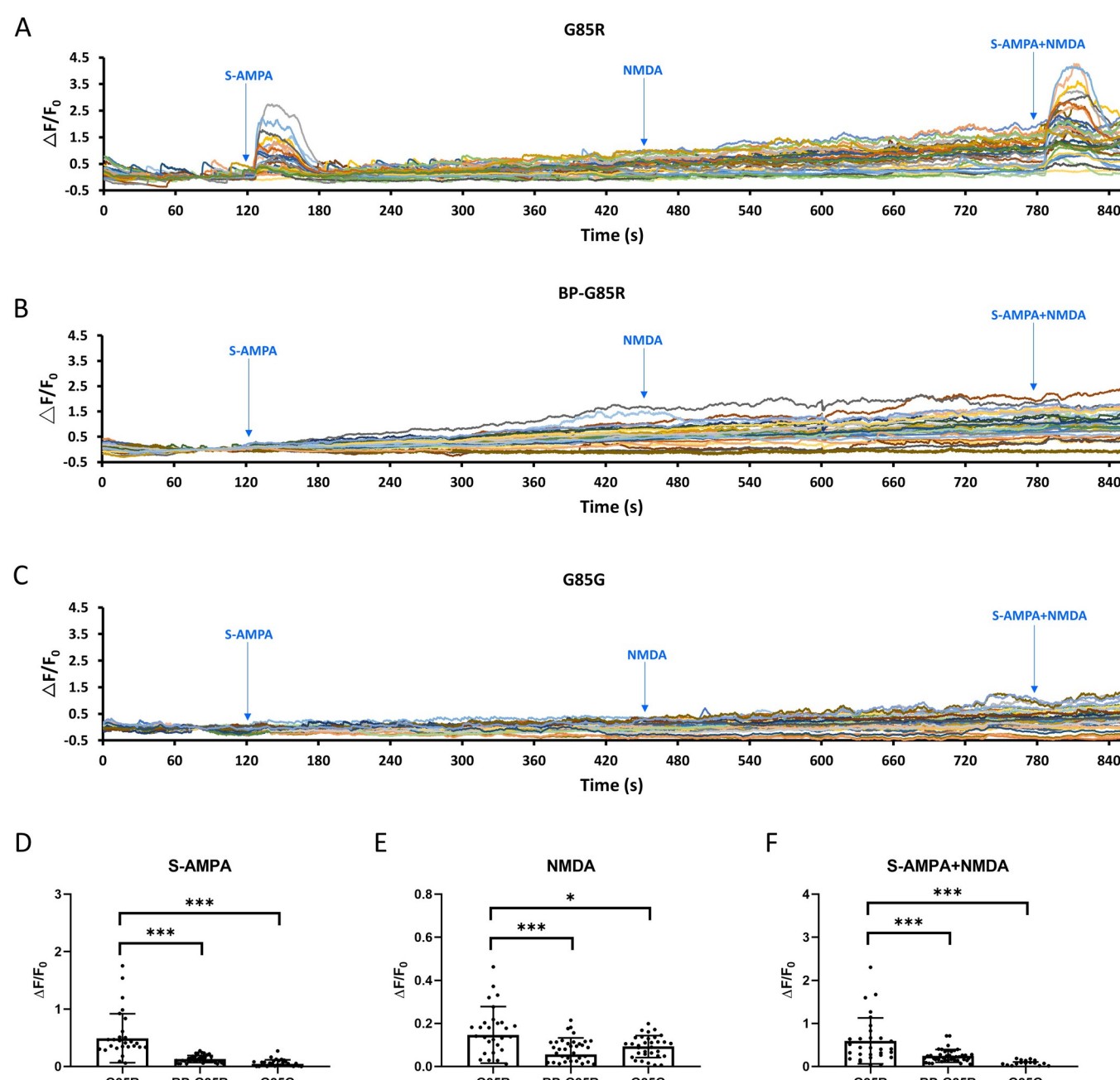

**Fig 4. Treatment with BP can effectively attenuate excessive calcium influx triggered by S-AMPA and NMDA in ALS-SOD1 mutant MNs.** (A, B and C) Microfluorometric measurements of intracellular calcium transients in untreated SOD1$^{G85R}$ iPSC-derived MNs (A), SOD1$^{G85R}$ iPSC-derived MNs treated with 10 μM BP (B), and SOD1$^{G85G}$ iPSC-derived MNs (controls) (C) loaded with Fluo-4 and exposed to S-AMPA at 120 s, NMDA at 450 s and S-AMPA + NMDA at 780 s. (D, E and F) Average △F/F$_0$ ratios during peak calcium induced by S-AMPA (D), NMDA (E) and S-AMPA + 40 μM NMDA (F). n = 30, 41 and 32 MNs for G85R, BP-G85R and G85G group, respectively. * p < 0.05 and ***p < 0.005 vs. the untreated SOD1$^{G85R}$ group.

returning to a level similar to that of SOD1$^{G85G}$ MNs (S-AMPA: 0.130 ± 0.0617; NMDA: 0.055 ± 0.0773; S-AMPA+NMDA: 0.247 ± 0.1486) (Fig 4B). Calculations of the calcium flux induced by S-AMPA, NMDA, and S-AMPA + NMDA in all groups were presented in Fig 4D–4F. The calcium flux induced by S-AMPA (Fig 4D), NMDA (Fig 4E), and S-AMPA + NMDA (Fig 4F) were significantly lower in SOD1$^{G85G}$ MNs compared to that of SOD1$^{G85R}$ MNs. It

was further revealed that BP treatment significantly reduced the calcium flux in SOD1$^{G85R}$ MNs, bringing it to a level comparable to that of SOD1$^{G85G}$ MNs. Additionally, BP was found to be more effective at inhibiting excessive S-AMPA-induced calcium influx compared to NMDA-induced calcium influx (Fig 4D and 4E). Collectively, our Western blot and calcium imaging results confirmed that BP restored normal calcium influx by inhibiting the expression of AMPAR and NMDAR in SOD1 mutant MNs.

## BP inhibited cell apoptosis and attenuated neurite degeneration in SOD1 mutant MNs

Accumulation of misfolded proteins and ensuing structural damage to neurites and axons is a pathological hallmark of ALS [10,46]. To study whether BP can diminish neurite degeneration and reduce motor neurons loss, SOD1$^{G85R}$ and SOD1$^{G85G}$ MNs treated with 10 μM BP for 72 h were immunostained for NF-H to present their neurite areas. The NF-H expression rate of control SOD1$^{G85G}$ MNs was 100%, whereas that of SOD1$^{G85R}$ MNs was merely 55.91% (Fig 5B). However, BP treatment increased this ratio to 84.62%, and neurite morphology was more robust than that in untreated SOD1$^{G85R}$ MNs (Fig 5A and 5B), revealing that BP can inhibit NF degeneration associated with the SOD1$^{G85R}$ mutation. The expression levels of activated caspase-3 were examined because it has been reported that this apoptosis effector protease is strongly associated with MN loss and damage due to ALS-related SOD1 mutations in cultured

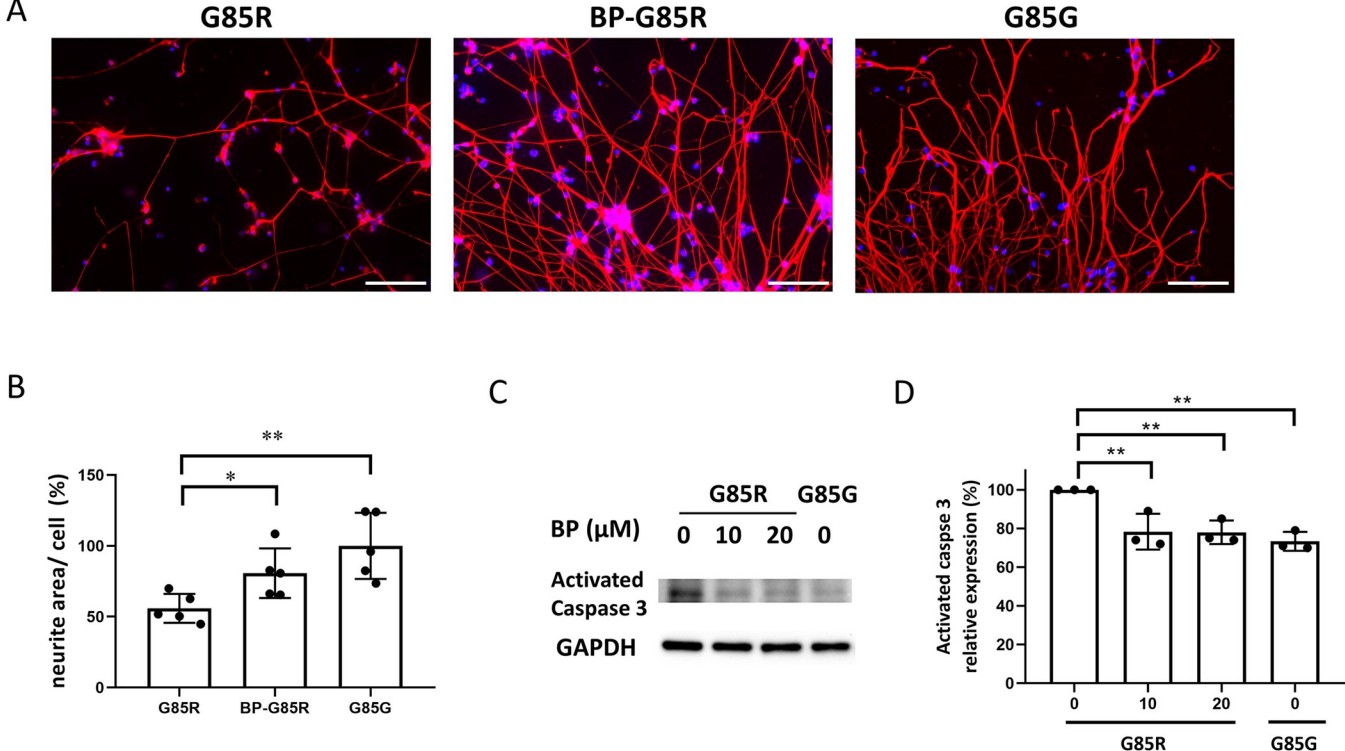

**Fig 5. Treatment with BP attenuates neurite degeneration and apoptosis in MNs harboring the ALS-associated SOD1$^{G85R}$ mutation.** (A) Neurite density of untreated SOD1$^{G85R}$ iPSC-derived MNs (left panel), SOD1$^{G85R}$ iPSC-derived MNs treated with 10 μM BP (middle panel), and genetically corrected SOD1$^{G85G}$ iPSC-derived MNs (right panel) as revealed by NF-H immunofluorescence staining (typical results from n = 5 cultures per group). Scale bar = 200 μm. (B) Quantification of neurite area/cell based on NF-H immunofluorescence staining using ImageJ; 60 to 195 cells per view were counted. * p < 0.05 and ** p < 0.01 vs. the untreated SOD1$^{G85R}$ group. (C and D) Western blotting for the apoptosis marker activated (cleaved) caspase-3. Expression levels in SOD1$^{G85R}$, BP treatment (10 and 20 μM), and SOD1$^{G85G}$ (control) groups in iPSC-derived MNs were quantified using ImageJ.

mouse spinal cord MNs and iPSC models [1,47,48]. Expression levels of activated caspase-3 was 27% higher in untreated ALS-SOD1$^{G85R}$ MNs compared to SOD1$^{G85G}$ MNs, but was reduced almost to the level of the SOD1$^{G85G}$ group by BP treatment (Fig 5C and 5D). Thus, we demonstrated that BP prevented MN death in the iPSC model of ALS, possibly by blocking the apoptosis pathway.

## BP could activate autophagy in SOD1 mutant MNs

Calcium homeostasis influences apoptosis by regulating various proteolytic processes as well as autophagy, which is a biologically conserved pathway for the elimination of damaged cellular macromolecules and recycling of the constituents [49]. Therefore, we investigated whether BP can reduce MNs loss and restore calcium homeostasis related to autophagic pathway using the human iPSCs model. Expression of the early stage autophagy marker LC3BII was substantially lower in untreated ALS-SOD1$^{G85R}$ MNs compared to SOD1$^{G85G}$ MNs but was enhanced to the level of SOD1$^{G85G}$ MNs by BP (Fig 6A and 6B). Conversely, expression of the late stage autophagy marker p62 was substantially higher in SOD1$^{G85R}$ MNs than SOD1$^{G85G}$ MNs, but again was reduced to near SOD1$^{G85G}$ MN levels by 20 μM BP (Fig 6C and 6D). Collectively, our results indicated that the autophagy pathway was induced after BP treatment, which further resulted in the prevention of apoptotic MN death in SOD1$^{G85R}$ MNs.

## Discussion

Dysregulation of intracellular calcium homeostasis is widely regarded as a key pathomechanism common to multiple neurodegenerative diseases, including ALS [44]. A sustained elevation in intracellular calcium concentration can induce endoplasmic reticulum (ER) stress and disruption of post-translational protein processing, failure of mitochondrial ATP generation, activation of catabolic enzymes that damage cellular macromolecules, and generation of ROS leading to oxidative stress [16,24,27,50,51]. Preventing unregulated calcium increases or restoring cellular calcium homeostasis is thus a promising therapeutic strategy. Here we demonstrate that BP treatment can reduce glutamate-induced calcium influx in ALS patient-derived MNs, likely by reversing overexpression of NMDA NMDAR1 and AMPA GluR3 subunits. Similarly, BP was reported to reduce activities of the calcium-activated proteases calpastatin and calpain in a SCA model, leading to a reduction in toxic protein fragment formation and ensuing neurotoxicity [36,52]. Thus, treatment with BP may be generally effective for restoring neuronal calcium homeostasis in neurodegenerative diseases of distinct etiology.

In contrast to this reduction in glutamate-induced calcium influx, BP enhanced KCl-induced influx. Superfusion of high-KCl is commonly used in physiological research to enhance membrane excitability, promote nerve conduction [53], and activate voltage-gated calcium channels (VGCCs) [54]. We suggest that this restoration of VGCC-mediated influx may also contribute to a therapeutic effect by restoring normal membrane excitability and synaptic communication.

Based on evidence for excitotoxicity in neurodegenerative diseases, several glutamate receptor antagonists have been developed as potential therapeutics. For example, the competitive NMDAR antagonist Riluzole is FDA-approved for reducing neuronal excitotoxicity in ALS. In addition to inhibiting NMDAR overexpression and potential hyperactivity; however, BP can also reduce the expression of AMPARs, ionotropic receptor-channels that can both enhance neuronal excitability and in some molecular conformations directly mediate calcium influx. Therefore, BP may act as a multimodal protective agent against calcium overload and neuronal excitotoxicity.

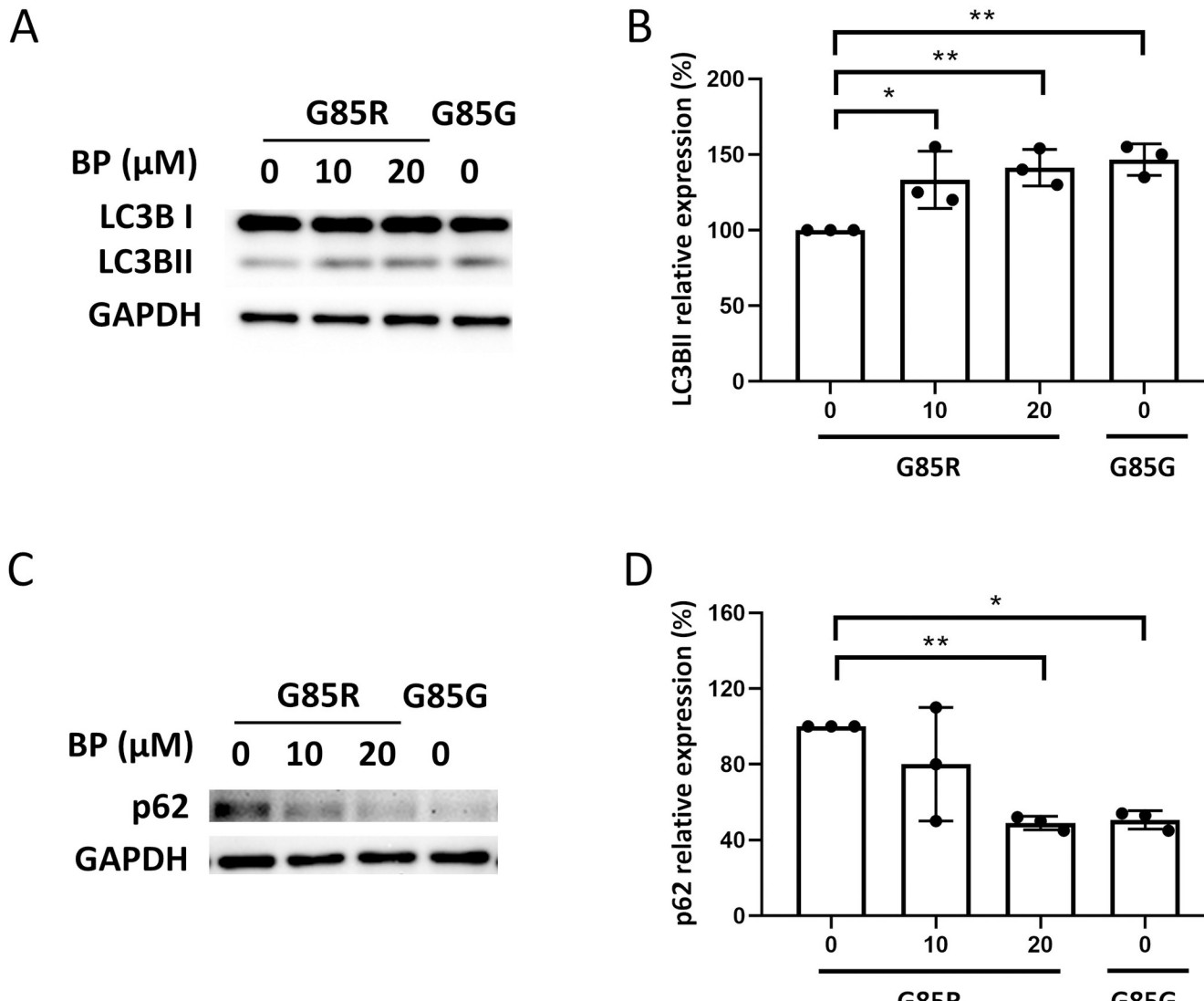

**Fig 6. Treatment with BP activates the autophagic pathway in ALS-SOD1 mutant MNs.** (A and C) Western blots of the early autophagy marker LC3BII (A) and late-stage marker p62 (C) from untreated SOD1$^{G85R}$ iPSC-derived MNs, SOD1$^{G85R}$ iPSC-derived MNs treated with BP (10 and 20 μM), and SOD1$^{G85G}$ iPSC-derived MNs. (B and D) Quantification of the signal was performed using ImageJ.

In addition to regulating the expression of glutamate receptors, BP enhanced the autophagic pathway (Fig 5), consistent with the BP treatment response reported in a SCA-iPSC model and with these studies suggesting that BP enhances autophagy and influences autophagosome formation [37]. However, the autophagic pathway was inhibited by BP treatment in a SOD1$^{G93A}$ mutant mouse model of ALS, and the effect actually prolonged mouse survival [34]. This discrepancy may be explained by different mutations and distinct downstream pathogenic mechanisms. Also, it is speculated that gene copy number varies between iPSCs and animal models, again leading to distinct pathogenesis [40,41]. Compared to the MNs derived from normal iPSCs, down-regulation of autophagy markers was observed in the MNs derived from C9ORF72 mutated iPSCs [55]. However, compared to WT mice, up-regulation of autophagy markers was observed in C9ORF72 Transgenic Mice [56]. These discrepancies may reflect differences in the duration of drug treatment. For example, the antihistaminergic

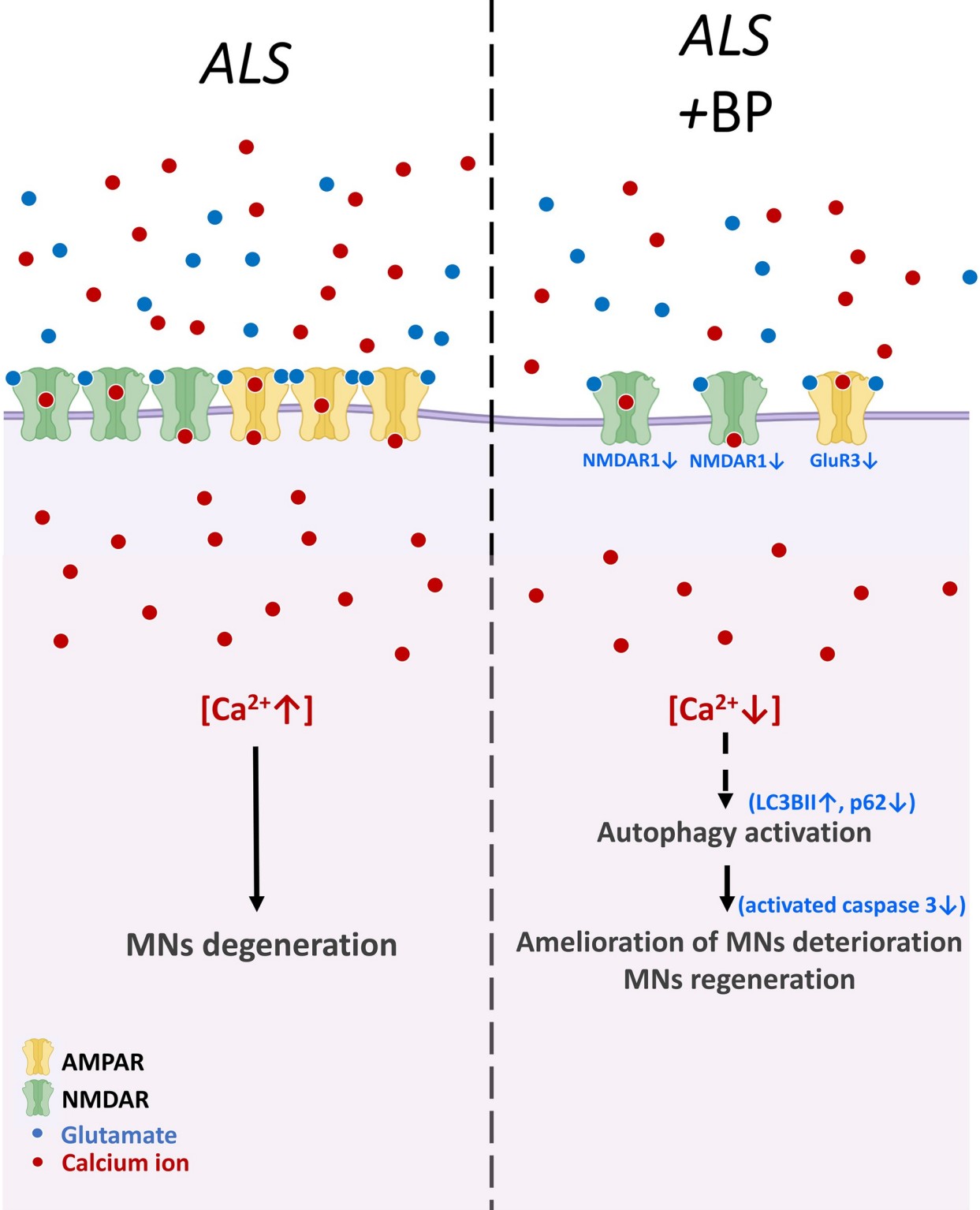

**Fig 7. Schematic illustrating the mechanisms by which BP treatment may reduce glutamate excitotoxicity in SOD1$^{G85R}$ motor neurons.** In ALS, excess glutamate release overstimulates calcium-permeable postsynaptic AMPARs and NMDARs. In addition, reversal of the glutamate transporter adds to this synaptic glutamate and stimulates further calcium influx into postsynaptic neurons. Calcium overload induces cell damage and eventually apoptosis and neuronal degeneration. The results of this study suggest that BP can reverse GluR3 and NR1 overexpression, thereby reducing excessive calcium accumulation in postsynaptic motor neurons. Furthermore, BP can also activate the autophagic pathway to prevent MN apoptosis.

clemastine differentially regulated autophagy in an ALS model depending on duration [57], with short-term treatment increasing LC3BII marker expression but long-term administration reducing LC3BII expression. In the current study as well, a 72-h treated enhanced LC3BII expression (Fig 5A) while a 3-h treatment was found to inhibit LC3BII expression in the SOD1$^{G93A}$ mutant mouse model of ALS [34]. The duration of BP treatment may thus differentially influence the expression of other proteins, potentially including VGCCs and glutamate receptor subunits. Additional studies are needed to examine the effects of different BP doses and durations on various ALS models.

Although our iPSC-based ALS model provided a precise representation that closely mimics real patient conditions, and we successfully identified potential compounds and related mechanisms for ALS therapy, there are several limitations to this study. First, we did not conduct extensive testing of BP's effective concentration and treatment duration. However, BP has been previously applied in neurological models, including both animal and cell models [29,34–37], which may allow us to predict the potential effective concentration and treatment duration in our model based on prior reports. The second limitation lies in the strategies used to uncover the potential mechanisms of BP in ALS. In this study, we did not employ systematic approaches such as RNA sequencing or proteomic analysis; therefore, the detailed mechanisms of BP remain to be further explored in future research. Additionally, our ALS motor neuron (MN) model has its own limitations. We utilized G85R SOD1 mutant MNs for further studies, which restricts the findings to a specific mutant type of familial ALS. The applicability of our findings to multiple genetic mutations and sporadic ALS needs further investigation. Moreover, our model is a simplified, single-cell type model that lacks interactions between different cell types, such as glial cells, muscles, and immune cells. Therefore, a comprehensive 3D organoid or animal model is necessary to evaluate the clinical potential of our findings.

In the current model, increased autophagy may have contributed to the improvement in calcium homeostasis and the reduction in apoptosis signaling. In addition, BP increased the neurite density of SOD1$^{G85R}$ iPSC-derived MNs compared to the untreated group, suggesting potential benefits for information processing. Previous studies have suggested that BP may stimulate neurogenesis [37], and Chi and coworkers (2018) reported that BP increased the neural differentiation of adipose-derived stem cells [58]. We suggest that BP may protect neurons both by preventing apoptosis and by promoting nerve regeneration from progenitors.

## Conclusion

As summarized in Fig 7, we demonstrated that BP can prevent or reverse neuron degeneration, reduce the overexpression of glutamate receptor subunits GluR3 and NMDAR1, inhibit excessive glutamate-induced calcium influx, and prohibit hyperactivated apoptosis signaling in motor neurons derived from ALS patient iPSCs harboring the SOD1$^{G85R}$ mutation. Moreover, enhanced autophagy may contribute to these effects, although further experimental verification is required. These findings highlight the therapeutic potential of BP for ALS and the utility of this experimental system as a drug-screening platform for neurodegenerative diseases.

## Supporting information

**S1 Fig. Original blot images in this study.**
(PDF)

**S2 Fig. Original immunofluorescent images in this study.**
(PDF)

**S1 File.**
(XLSX)

**S1 Raw images.**
(PDF)

## Author Contributions

**Conceptualization:** Yu-Chen Deng, Horng-Jyh Harn, Chia-Yu Chang, Tzyy-Wen Chiou.

**Data curation:** Yu-Chen Deng.

**Formal analysis:** Yu-Chen Deng, Jen-Wei Liu, Chia-Hung Chiang.

**Funding acquisition:** Chia-Yu Chang, Tzyy-Wen Chiou.

**Investigation:** Yu-Chen Deng, Tzu-Chen Kuo, Chia-Hung Chiang, En-Yi Lin.

**Methodology:** Jen-Wei Liu, Hsiao-Chien Ting, En-Yi Lin.

**Resources:** Shinn-Zong Lin, Chia-Yu Chang, Tzyy-Wen Chiou.

**Supervision:** Horng-Jyh Harn, Chia-Yu Chang, Tzyy-Wen Chiou.

**Validation:** Yu-Chen Deng, Tzu-Chen Kuo.

**Visualization:** Tzu-Chen Kuo.

**Writing – original draft:** Yu-Chen Deng, Jen-Wei Liu.

**Writing – review & editing:** Yu-Chen Deng, Horng-Jyh Harn, Chia-Yu Chang, Tzyy-Wen Chiou.

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
