## [Decision Letter · Decision Letter 0]

14 Jun 2024

PONE-D-24-19706n-Butylidenephthalide recovered calcium homeostasis to ameliorate neurodegeneration of motor neurons derived from amyotrophic lateral sclerosis iPSCsPLOS ONE

Dear Dr. Deng,

Thank you for submitting your manuscript to PLOS ONE. After careful consideration, we feel that it has merit but does not fully meet PLOS ONE’s publication criteria as it currently stands. Therefore, we invite you to submit a revised version of the manuscript that addresses the points raised during the review process.

After careful consideration by 2 Reviewers and an Academic Editor, all of the critiques of both Reviewers must be addressed in detail in a revision to determine publication status. If you are prepared to undertake the work required, I would be pleased to reconsider my decision, but revision of the original submission without directly addressing the critiques of the Reviewers does not guarantee acceptance for publication in PLOS ONE. If the authors do not feel that the queries can be addressed, please consider submitting to another publication medium. A revised submission will be sent out for re-review. The authors are urged to have the manuscript given a hard copyedit for syntax and grammar.

We look forward to receiving your revised manuscript.

Kind regards,

Stephen D. Ginsberg

Section Editor

PLOS ONE

Journal Requirements:

'The authors are grateful to the funding from National Science and Technology Council in Taiwan (MOST 111-2314-B-303-027-MY2) and the support from the core facilities provided by the Instrument Center of Department of Medicine Research, Buddhist Tzu-Chi General Hospital and Bioinnovation Center, Buddhist Tzu Chi Medical Foundation, Hualien, Taiwan.'

'Funder: National Science and Technology Council

Award Number: MOST 111-2314-B-303-027-MY2

Recipient: Chia-Yu Chang

This funder provides study design, data collection, and analysis. They also make decisions regarding publication.'

Reviewers' comments:

Reviewer's Responses to Questions

**Comments to the Author**

1. Is the manuscript technically sound, and do the data support the conclusions?

Reviewer #1: Partly

Reviewer #2: Yes

2. Has the statistical analysis been performed appropriately and rigorously? 

Reviewer #1: N/A

Reviewer #2: Yes

3. Have the authors made all data underlying the findings in their manuscript fully available?

Reviewer #1: Yes

Reviewer #2: Yes

4. Is the manuscript presented in an intelligible fashion and written in standard English?

Reviewer #1: Yes

Reviewer #2: Yes

5. Review Comments to the Author

Reviewer #1: The authors follow an interesting strategy and show possible targets for a neurodegenerative disease such as ALS, for which there is currently no cure.The structure is very easy to understand and clearly arranged with a very nice summarizing figure 6, even if the introduction almost fails to mention more recent publications/studies, especially in the context of ALS mutations and Ca-imaging. (e.g. PMID 37595581 or PMID 31105804).

Furthermore, the authors should revise the results and discussion section again and make sure that there is not already too much discussion in the results section. Besides that the limitations of this study should be mentioned in the discussion, especially with regard to the treatment or the treatment concentrations/ treatment time, cell culture effects, etc.

In addition, clarification is needed on the experiments carried out, particularly in relation to the n-number:

1.) What does the sentence in line 321 “Mean ± SD of n = 4 independently treated cultures per treatment group” mean?

Normally at least 2-3 biological replicates with 2-3 technical replicates should be made in the cell culture to exclude possible passages/aging effects in the cell culture. Thus, at least 2+ different passages should have been differentiated to MNs.

In addition, where do the concentrations of the BP treatment come from, has an MTT-/LDH-assay been carried out with regard to the tolerability of the treatment to figure out the best concentration.

2.) The number of MNs tested in calcium imaging with only 9 individual MNs per group is too low. Under "normal lab" conditions, this is just the minimum number of cells that should be examined in a single Ca imaging measurement. Therefore, more MNs would be necessary, e.g. through further technical replicates AND biological through further differentiations.

 It would have been nice to show the possibly different Ca baselines between the lines first, especially with the isogenic control line (=analysis of basal intracellular Ca2+ levels and spontaneous Ca2+ signals is missing).

 Furthermore, the authors discuss several times that treatment with BP also has an effect on different receptors. The question then arises as to why this was not also tested with S-AMPA or NDMA, for example, in order to detect their effects on the release of Ca.

3.) Figures 3-5 lack information on the amount of loaded protein and the n-number of analyses for all Western blots.

If the blot was only done once, where do the error bars in the graphs come from?

If several blots were made, then point out to the reader that the blot shown is a “representative blot” and show all blots for the respective figures in the Supp section.

Finally, all graphs should be displayed as box and whiskers plots including “show all points” to make the results clearer for the reader.

Reviewer #2: In the manuscript “n-Butylidenephthalide recovered calcium homeostasis to ameliorate neurodegeneration of motor neurons derived from amyotrophic lateral sclerosis iPSCs” the authors tested the protective efficacy of n-Butylidenephthalide (BP) on the survival of motor neurons (MN) derived from patients with Amyotrophic lateral sclerosis (ALS). They found that BP treatment restored the “normal” channels function and activated autophagic pathways that attenuated neuron degeneration. They suggest that BP is a promising candidate for future treatment of ALS.

While the data is well-organized, and experiments were conducted properly, there are some concerns that must be addressed before is achieved a suitable version of the manuscript for publication.

Main concern

The scheme depicted in figure 6 does not clearly illustrate the obtained results and include elements not studied in the research performed (i.e. NCX, PMCA). Furthermore, no synapse-specific tests were performed in the conducted experiments, all data is related to “post-synaptic” effects. Please clarify the concepts outlined in the figure and redraw the diagram.

Minor comments

Line 68 mistyped word: “antagonized” change to antagonize.

Figure 2. It would be helpful to add the cell type in labels where the BP was applied (i.e. BP-G85R)

6. PLOS authors have the option to publish the peer review history of their article (what does this mean?). If published, this will include your full peer review and any attached files.

Reviewer #1: No

Reviewer #2: **Yes: **Adán Dagnino Acosta

---

## [Author Response · Author response to Decision Letter 0]

13 Sep 2024

Dear reviewers,

We followed Oriel Jerome Delas Alas Vida’s suggestion and incorporated raw data, and shared the "minimal data set" for the manuscript to reflect the suggestions that provided on 8/28.

The content includes: 

Figure 1E: Original immunofluorescent images and quantified data of HB9/ DAPI.

Figure 3B (NMDAR1), 3D (GluR3), 5D (activated caspase 3), 6B (LC3BII), 6D (p62): Original blotting images and quantified data of Western Blot.

Figure 5B: Original immunofluorescent images and quantified data of neurofilament/ DAPI. 

Figure 2C: The quantified data of basal calcium content.

Figure 2G, 2H: The quantified data of G85R, BP-G85R, and G85G under KCl and L-glutamate explored.

Figure 4D, 4E, 4F: The quantified data of G85R, BP-G85R, and G85G under agonist (S-AMPA, NMDA and S-AMPA+NMDA).

We appreciate the time and effort that you have dedicated to providing insightful feedback to strengthen our paper. The following is a point-by-point response to the questions and comments has been raised.

Comments from reviewers and the responses:

Reviewer 1:

Major revisions

1. The authors follow an interesting strategy and show possible targets for a neurodegenerative disease such as ALS, for which there is currently no cure. The structure is very easy to understand and clearly arranged with a very nice summarizing figure 6, even if the introduction almost fails to mention more recent publications/studies, especially in the context of ALS mutations and Ca-imaging. (e.g. PMID 37595581 or PMID 31105804).

[RESPONSE]: 

Thank you very much for reminding us and making our content better. The introduction has been updated to include the recent publications. In response to your suggestion, PMID 37595581 was included to introduce that ALS-associated mutations exhibit impaired calcium homeostasis, leading to changes in excitability, failure of action potential conduction, and activation of various calcium-dependent degenerative pathways (Lines 51 (Yang et al., 2023 (14)). However, PMID 31105804 primarily discusses the Thrifty Food Plan recommendations, which do not appear to be related to ALS or motor neurons. Additionally, we reviewed the publications by the authors of that article but were unable to find any that are related to ALS mutations and calcium imaging. Therefore, a reference reviewing the excitotoxicity caused by calcium dysregulation in ALS has been cited instead (Line 51, Verma et al., 2022 (15); PMID: 35078537).

2. Furthermore, the authors should revise the results and discussion section again and make sure that there is not already too much discussion in the results section.

[RESPONSE]:

Thank you for your suggestions to improve our content and structure. Thank you for your suggestions to improve our content and structure. The results section has been revised to avoid excessive discussion (lines 172-174, 183-186, 202-203, 212-215, 222-224 of the original manuscript). Additionally, redundant introductory statements that were already provided in the introduction have been deleted. 

3. Besides that the limitations of this study should be mentioned in the discussion, especially with regard to the treatment or the treatment concentrations/ treatment time, cell culture effects, etc.

[RESPONSE]: 

Thank you for your suggestions to enrich our content and make it more comprehensive. We have addressed the following limitations in the discussion section:

Drug’s concentration and treatment duration: We did not conduct extensive testing of BP’s effective concentration and treatment duration. However, BP has been previously applied in neurological models, including both animal and cell models, which may allow us to predict the potential effective concentration and treatment duration in our model based on prior reports (Line 285-287). The results, including the effective dose (ranging from 10 to 530 μM) and the effective treatment duration (ranging from 6 to 72h) in cell models, were revealed in this study as expected. 

Comprehensive study on the mechanism: In this study, we focused on the mechanisms by which BP regulates calcium influx, calcium-associated receptors, and improves the survival of neuron cells. Systematic approaches, such as RNA sequencing or proteomic analysis, to further uncover the detailed mechanisms of BP remain to be explored in future research. (Lines 287-290).

Relevance of cell model: We utilized SOD1G85R mutant MNs as the cell model in this study, which limits the findings to a specific type of familial ALS. The applicability of our findings to multiple genetic mutations and sporadic ALS needs further investigation (Lines 290-292). Moreover, a comprehensive 3D organoid or animal model may better simulate the ALS disease condition than using a single cell type when studying the treatment potential of a drug, and this approach can be further improved in future studies. (Lines 292-294)

4. In addition, clarification is needed on the experiments carried out, particularly in relation to the n-number:

1.) What does the sentence in Line 321 “Mean ± SD of n = 4 independently treated cultures per treatment group” mean? Normally at least 2-3 biological replicates with 2-3 technical replicates should be made in the cell culture to exclude possible passages/aging effects in the cell culture. Thus, at least 2+ different passages should have been differentiated to MNs.

[RESPONSE]:

Thank you for your suggestions so that we can better clarify the content. Originally, two biological replicates and two technical replicates were performed to conduct statistical analysis. Although two different passages were tested in the original submitted version, we have included three biological replicates (G85R: passage numbers 40, 32 and 25; G85G: passage numbers 44, 35 and 29) with three technical replicates in the revised manuscript to mark our statements more convincing. The updates are on Figure 1E.

5. In addition, where do the concentrations of the BP treatment come from, has an MTT-/LDH-assay been carried out with regard to the tolerability of the treatment to figure out the best concentration.

[RESPONSE]: 

Thank you very much for your suggestion. The concentrations of BP treatment were based on a previously published iPSC model (Chang, 2014) from our group. MTT assays were used to determine the cytotoxicity of BP on DS-iPSC-derived neurons (concentrations tested: 0, 10, 50, and 100 μM). The results showed that 100 μM BP caused significant cell death in DS-iPSC-derived neurons, while 50 μM BP resulted in 80% cell viability. Based on these results, 10 μM and 20 μM were selected for our study. We added the reference on Line 155. 

6. 2.) The number of MNs tested in calcium imaging with only 9 individual MNs per group is too low. Under "normal lab" conditions, this is just the minimum number of cells that should be examined in a single Ca imaging measurement. Therefore, more MNs would be necessary, e.g. through further technical replicates AND biological through further differentiations.

[RESPONSE]:

Thank you for the constructive opinion. We have repeated the study to further improve this figure. MNs (D42-D46) were subjected to three replicate experiments, and the AMPAR and NMDAR agonists were introduced to present the overactive calcium influx situation. Additionally, Fluo-4 was used instead of Fura-2 because of the repair and replacement of the detection instruments. The number of evaluated MNs was also increased to n = 95, 55, and 49 for the G85R, BP-G85R, and G85G groups induced with KCl and L-glutamate, respectively, and to n = 30, 41, and 32 for the G85R, BP-G85R, and G85G groups induced with agonists, respectively. In brief, compared with the G85G group (control), the G85R group showed distinct difference when stimulated with KCl and L-glutamate, respectively. In G85R group, it showed a significant decrease in calcium influx (△F/F0) under KCl (0.518 ± 0.3356) and a greatly increase in calcium signaling under L-glutamate (3.130 ± 1.3304). Moreover, BP treated-group (KCl: 1.063 ± 1.4433; L-glutamate: -0.044 ± 0.3722) has the same trend as G85G (KCl: 1.258 ± 0.8873; L-glutamate: 0.265 ± 0.1904). In addition, stimulating three groups (G85R, BP-G85R and G85G) with glutamate receptor agonist also showed that G85R (S-AMPA: 0.491 ± 0.4262, NMDA: 0.147± 0.1312; S-AMPA + NMDA: 0.596 ± 0.5328) had a higher calcium signal, while BP-G85R (S-AMPA: 0.130 ± 0.0617; NMDA: 0.055 ± 0.0773; S-AMPA+NMDA: 0.247 ± 0.1486) and G85G (S-AMPA: 0.044 ± 0.0724; NMDA: 0.093 ± 0.0508; S-AMPA+NMDA: 0.017 ± 0.0888) had a significant downward trend. These findings suggest that BP treatment can restore voltage-gated influx and prevent glutamate related transmitter-induced excitotoxicity in MNs carrying an ALS-associated SOD1 mutation to a healthy state. These results are presented in Figures 2G, 2H, 4D, 4E, and 4F, and are described in the figure legends and in the manuscript (Lines 189 to 199; 212 to 222).

7. It would have been nice to show the possibly different Ca baseLines between the Lines first, especially with the isogenic control Line (=analysis of basal intracellular Ca2+ levels and spontaneous Ca2+ signals is missing).

[RESPONSE]:

Thank you very much for pointing out what we need to improve to make our content more complete. The figures have been revised to present the Ca2+ baselines (Figure 2C) and spontaneous Ca2+ signals (Figure 2A and 2B) at the beginning in G85R and G85G group. The results showed no significant difference in basal intracellular calcium levels and spontaneous calcium transients of MNs among these two groups (Fig 2A, 2B and 2C, G85R: 0.196 ± 0.1251, G85G: 0.154 ± 0.1035). The conclusion is described in Line 185 to 187.

8. Furthermore, the authors discuss several times that treatment with BP also has an effect on different receptors. The question then arises as to why this was not also tested with S-AMPA or NDMA, for example, in order to detect their effects on the release of Ca.

[RESPONSE]:

Thank you for your suggestions to make our content more complete. To assess the impact of BP on different glutamate receptors, we utilized specific agonists for NMDA and AMPA (S-AMPA, NMDA, S-AMPA+NMDA) to stimulate three groups of cells (SOD1G85R, SOD1G85R+BP, SOD1G85G; Figure 4A, 4B and 4C). The results, as shown in Figures 4A, 4B, and 4C, with statistical analysis of the ΔF/F0 ratio presented in Figure 4D (S-AMPA), 4E (NMDA), and 4F (S-AMPA+NMDA). Our calcium image results reflected the findings from the Western blot results in Figure 3. These data indicated that BP not only reduces the abnormal overexpression of NMDAR1 and GluR3 in SOD1G85R MNs but also decreases the calcium flux in response to NMDA and AMPA-specific agonists. The description of these results is provided in Lines 208 to 224.

9. Figures 3-5 lack information on the amount of loaded protein and the n-number of analyses for all Western blots. If the blot was only done once, where do the error bars in the graphs come from? If several blots were made, then point out to the reader that the blot shown is a “representative blot” and show all blots for the respective figures in the Supp section.

[RESPONSE]:

Thank you very much for your reminder to improve our content. We have updated the original data of triplicate blotting in Figures 3-5 (have been convert to Figure 3, 5 and 6) in the supporting data. We have meticulously and thoroughly re-identified all the data. The results of the analysis do not affect the statistical trend, and we have also confirmed their accuracy. 

10. Finally, all graphs should be displayed as box and whiskers plots including “show all points” to make the results clearer for the reader. 

[RESPONSE]:

Thank you very much for your reminder to make our content more complete and clearer. In fig 1E, 2C, 2G, 2H, 3B, 3D, 4D, 4E, 4F, 5B, 5D, 6B, 6D, we have displayed the data as box and whisker plots to enhance clarity and credibility.

Reviewer 2:

Major revisions

1. In the manuscript “n-Butylidenephthalide recovered calcium homeostasis to ameliorate neurodegeneration of motor neurons derived from amyotrophic lateral sclerosis iPSCs” the authors tested the protective efficacy of n-Butylidenephthalide (BP) on the survival of motor neurons (MN) derived from patients with Amyotrophic lateral sclerosis (ALS). They found that BP treatment restored the “normal” channels function and activated autophagic pathways that attenuated neuron degeneration. They suggest that BP is a promising candidate for future treatment of ALS.

While the data is well-organized, and experiments were conducted properly, there are some concerns that must be addressed before is achieved a suitable version of the manuscript for publication. The scheme depicted in figure 6 does not clearly illustrate the obtained results and include elements not studied in the research performed (i.e. NCX, PMCA). Furthermore, no synapse-specific tests were performed in the conducted experiments, all data is related to “post-synaptic” effects. Please clarify the concepts outLined in the figure and redraw the diagram.

[RESPONSE]:

Thank you for your suggestions to make our content clearer and more complete. We have updated the concept of Figure 7 (Figure 6→Figure 7) and redrawn the diagram. Since all the data relates to post-synaptic effects, we removed the description of the pre-synaptic gateway. In addition, we deleted the description of calcium-related channels such as PMCA, NCX, and VGCC, and focused on AMPAR and NMDAR, which are the main subjects of this article.

Minor revisions

2. Line 68 mistyped word: “antagonized” change to antagonize.

[RESPONSE]:

Thank you very much for your advice, we have made the corrections accordingly (Line 69). 

3. Figure 2. It would be helpful to add the cell type in labels where the BP was applied (i.e. BP-G85R)

 [RESPONSE]:

Thank you for your suggestions to make the article clearer and easier to understand. We have replaced the BP conditions in the figure with BP-G85R (see Figure 2E, 2G, 2H, 4B, 4D, 4E, 4F, 5A, 5B for details).

---

## [Decision Letter · Decision Letter 1]

23 Sep 2024

n-Butylidenephthalide recovered calcium homeostasis to ameliorate neurodegeneration of motor neurons derived from amyotrophic lateral sclerosis iPSCs

PONE-D-24-19706R1

Dear Dr. Deng,

We’re pleased to inform you that your manuscript has been judged scientifically suitable for publication and will be formally accepted for publication once it meets all outstanding technical requirements.

Kind regards,

Stephen D. Ginsberg, Ph.D.

Section Editor

PLOS ONE

**Comments to the Author**

1. If the authors have adequately addressed your comments raised in a previous round of review and you feel that this manuscript is now acceptable for publication, you may indicate that here to bypass the “Comments to the Author” section, enter your conflict of interest statement in the “Confidential to Editor” section, and submit your "Accept" recommendation.

Reviewer #1: All comments have been addressed

Reviewer #2: All comments have been addressed

2. Is the manuscript technically sound, and do the data support the conclusions?

Reviewer #1: Yes

Reviewer #2: Yes

3. Has the statistical analysis been performed appropriately and rigorously? 

Reviewer #1: Yes

Reviewer #2: Yes

4. Have the authors made all data underlying the findings in their manuscript fully available?

Reviewer #1: Yes

Reviewer #2: Yes

5. Is the manuscript presented in an intelligible fashion and written in standard English?

Reviewer #1: Yes

Reviewer #2: Yes

6. Review Comments to the Author

Reviewer #1: The authors answered and revised all the points raised to my complete satisfaction and also made further improvements to validate their data. Besides that die authors revised the content of their manuscript, so that it now reads much better and feels more coherent overall. Sorry for the error in my part in citing the wrong PMID number.

Reviewer #2: The authors have thoroughly reviewed and significantly enhanced the manuscript. Every point raised during the initial round of the review process has been carefully addressed and incorporated.

7. PLOS authors have the option to publish the peer review history of their article (what does this mean?). If published, this will include your full peer review and any attached files.

Reviewer #1: No

Reviewer #2: No

---

## [Editor Report · Acceptance letter]

29 Oct 2024

PONE-D-24-19706R1 

PLOS ONE

Dear Dr. Deng, 

I'm pleased to inform you that your manuscript has been deemed suitable for publication in PLOS ONE. Congratulations! Your manuscript is now being handed over to our production team.

Kind regards, 

on behalf of

Dr. Stephen D. Ginsberg 

Section Editor

PLOS ONE